# Band-Gap Engineering: Lithium Effect on the Electronic Properties of Hydrogenated 3C-SiC (1 1 0) Surfaces

Jose Luis Cuevas [1] , Miguel Ojeda Martinez [2] and Saravana Prakash Thirumuruganandham [1,*]

1   Centro de Investigación de Ciencias Humanas y de la Educación (CICHE), Universidad Indoamérica, Ambato 180103, Ecuador
2   Centro de Investigación en Nanociencia y Nanotecnología de CUValles, Departamento de Ciencias Naturales y Exactas, Universidad de Guadalajara, Ameca 46600, Jalisco, Mexico
*   Correspondence: saravprak@googlemail.com or saravanaprakash@uti.edu.ec; Tel.: +593-994765516

**Abstract:** Silicon carbide has structural strength, high electronic conductivity, low diffusion barrier and high storage capacity, which are suitable for engineering applications such as lithium-ion batteries, electric vehicles, uninterruptible power supplies and SiC diodes. In particular, 3C-SiC monolayers oriented along the (1 1 0) crystallographic direction that could have symmetric surfaces have been poorly studied, as have the effects of surface passivation on their physical and electronic properties. In this work, we investigate the influence of lithium on the electronic properties of hydrogenated surfaces in 3C-SiC monolayers using density functional theory. We examine the electronic properties of surfaces fully passivated with hydrogen with those of surfaces fully passivated with lithium and those with mixed passivation. Our results show that only fully hydrogenated surfaces exhibit a direct band-gap, while the full Li, CH+SiLi, and H+Li$_c$ passivations exhibit metallic behavior. The CLi+SiH, H+1LiC, and H+1LiSi passivation systems decrease the band-gap compared to the hydrogenated case and show an indirect band-gap. The formation energy of the system shows that the most stable arrangement is full-H, followed by H+1LiC, and the most unstable system is full-Li, which has a positive formation energy

**Keywords:** SiC–Li; lithium batteries; surfaces; DFT; electronic properties; band-gap; DOS; PDOS; GGA framework; formation energy



## 1. Introduction

Recently, great efforts have been made to develop new devices for energy storage [1–3]. One of the most researched storage devices is the lithium-ion battery (LIB). LIBs dominate the battery market for portable electronic devices such as laptops and cell phones. In addition, the LIB has also been successfully used as a new technology for automobiles such as hybrid (HEV), plug-in (PHEV), or all-electric vehicles (BEV), as well as for stationary energy storage [4,5]. In most LIBs, the commercially used anode material is graphite because it has a low and flat voltage range, high Coulomb efficiency, good cyclability, and low volume change upon Li insertion. However, one of the disadvantages of graphite is its comparatively low storage capacity. Graphite can store lithium with a capacity of one Li atom per six carbon atoms LiC$_6$, which corresponds to a specific electrical capacity of 372 mA h g$^{-1}$ [5–7]. A disadvantage during the process of charge and discharge is that the material tends to break producing a short lifetime of the battery [8,9]. To solve this situation some researchers have studied the improvement of the material with the use of 2D materials such as graphene, silicene, siloxene or germanene which have promising properties to be used as cathode in batteries [10–12] and also with the employment of other 2D nanoestructures as thin films or monolayers. Furthermore, Silicon Carbide (SiC) is a material that has been investigated due to special characteristics such as wide band-gap, and high thermal and chemical stability [13,14], especially because SiC is regarded to be electrochemically stable so it could be used as a buffer or matrix for lithium-ion

batteries [6,15,16]; in this sense, many studies have been performed about the employment of surfaces in 3C-SiC(100) monolayers [17–20], but there are only a few studies focused on 3C-SiC(1 1 0) layers. Theoretical and experimental studies have shown that clean 3C-SiC(1 1 0) surfaces have a metallic behavior in the band-gap [21–23]. As we know, some researchers focus on studying the different properties obtained by using (100), (111) and (1 1 0). In particular, Taehoon Park et al. [24] show that the (1 1 0) surface is the most sensitive to oxidation compared to the others, which is due to the number of silicon and carbon atoms that are on the surface because of the surface orientation. In this research, we studied the effects of lithium atoms on the hydrogenated 3C-SiC (1 1 0) surfaces. Our results show that only fully hydrogenated surfaces exhibit direct band-gap, while the passivations Full-Li, CH+SiLi and H+Li$_c$ have a metallic behavior. The passivations schemes CLi+SiH, H+1LiC and H+1LiSi reduce the band-gap compared with the hydrogenated case and exhibit an indirect band-gap. The formation energy of the system shows that the most stable disposition is the Full-H followed by H+1LiC, and the most unstable system is Full-Li which has positive formation energy.

## 2. Methods

In this work, the monolayers of 3C-SiC oriented in the crystallographic direction (1 1 0) were modeled by the supercell scheme of an otherwise perfect bulk SiC crystal [25]. The thickness of these monolayers is 3.07 Å, as shown in Figure 1. The monolayers are considered to be periodic in the $x$ and $y$ axes, while confined in the $z$ direction. To avoid the interaction between their replicas, we have considered a slap distance of more than 10 Å along the $z$-direction. The lattice parameters $a = 8.69$, $b = 6.14$, and $c = 18.07$ Å. In Figure 1a, we observed that the monolayers of Si and C atoms form on the top and bottom surfaces, which is why they are called symmetric monolayers. We have considered different passivation schemes over both surfaces to conform to the monolayer. First, we considered the bare case (Figure 1a), which is called pristine, then the surface was passivated with hydrogen atoms (Figure 1b), defined as full-H, leading to Si–H and C–H bonds. Then, the H atoms were replaced with lithium atoms, resulting in Si–Li and C–Li bonds (Figure 1c), which are referred to as Full-Li. To better understand the effects of lithium on the electronic properties, mixed passivations are considered, which consist of passivating the carbon atoms with hydrogen and the silicon with lithium (Figure 1d) and vice versa (Figure 1e). These passivations are referred to as CH+SiLi and CLi+SiH, respectively. Another mixed passivation consists of replacing only one H atom with Li in carbon or silicon atoms, respectively. These passivations are referred to as H+1LiC and H+1LiSi (Figure 1f,g). Finally, in the full-H case, we introduced only one lithium, which is denoted as H+Li$_c$. This lithium was placed in the center of the hexagon of the monolayer (Figure 1h). We used the density functional theory (DFT) described in the code SIESTA to perform all calculations [26]. The wave functions were represented with a double zeta basis set and polarization orbitals. The correlation of the electronic exchange energy was treated with the RPBE generalized gradient approximation (GGA).

An energy cutoff of 150 eV was used in the calculations. For the charge density integration we used an $8 \times 8 \times 1$ k-point Monkhorst Pack mesh. All atoms were allowed to relax until the forces were less than 0.001 eV/Å. The main parameters used in the calculations are summarized in Table 1. To ensure convergence of the results, we performed some convergence tests for the k-point network and the energy cutoff, both of which are shown in Supplementary Material S1 (see Figures S2 and S3 for details).

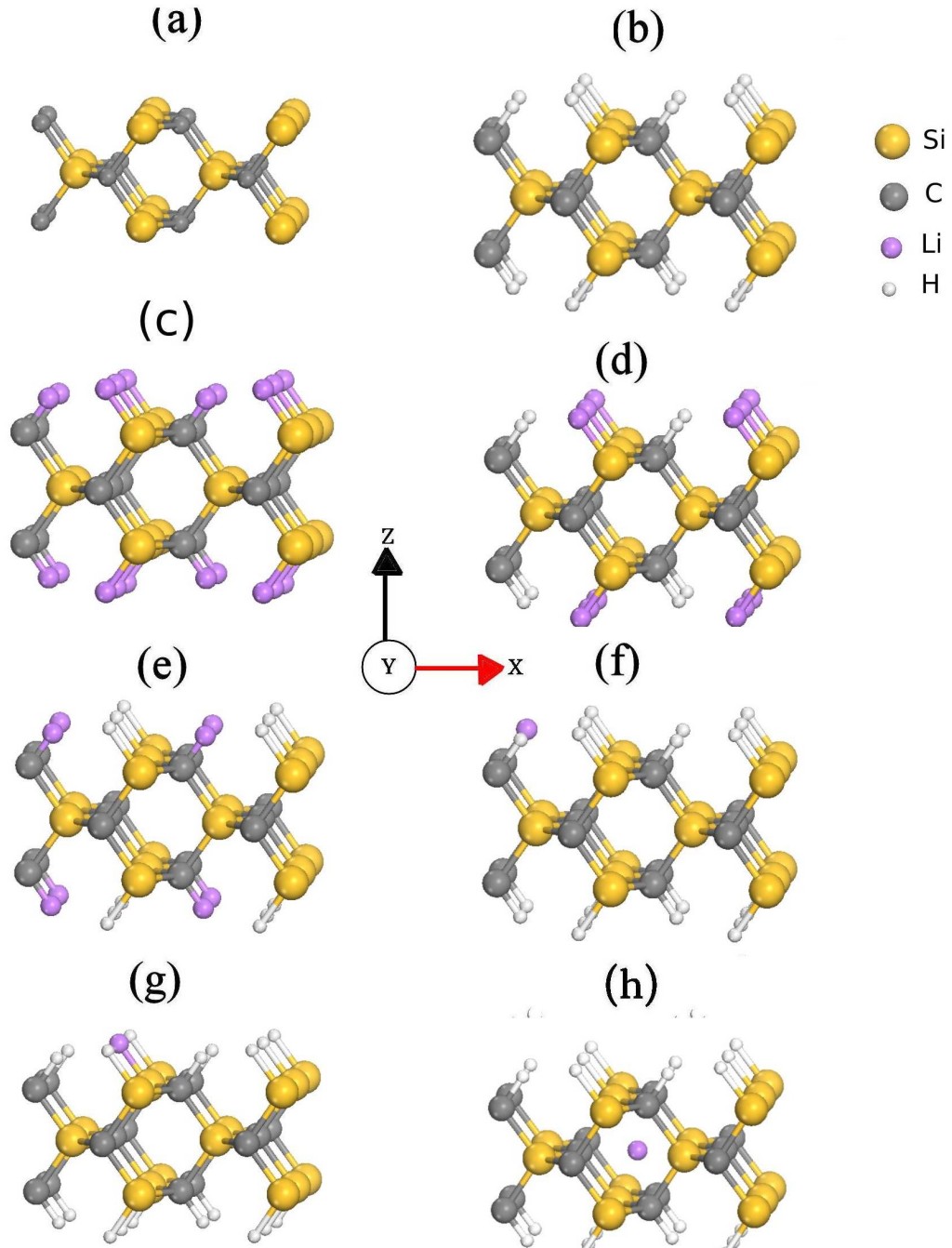

**Figure 1.** Schematic representation of 3C-SiC (1 1 0) monolayer with different passivations schemes: (**a**) Pristine; (**b**) Full-H; (**c**) Full-Li; (**d**) CH+SiLi; (**e**) CLi+SiH; (**f**) H+1LiC; (**g**) H+1LiSi; (**h**) H+Li$_c$. The grey, yellow, white and purple spheres represent the C, Si,H and Li atoms, respectively.

The formation energy ($E_f$) of the system was calculated according to the following formula [27]

$$E_f = \frac{E_0 - \sum_{i=\text{Zn,O,C,H,Li}} n_i \mu_i}{n_T} \tag{1}$$

Here, $E_0$ and $n_T$ stand for the ground state energy and the total number of atoms of the species per supercell and $n_i$, $\mu_i$ for the abundance of the chemical potentials. The chemical potentials for Si and C and Li were assumed to be the cohesive energies per atom of the crystal structure, while the H are the binding energies of the H$_2$ molecules, all calculated with the same level of theory. The energy formation was calculated using Equation (1).

First, we relaxed all SiC monolayers, considering in each case that the resulting value of the final energy corresponds to $E_0$. We have also added the values of binding energy per atom for the whole structure. This value represents the energy required to retrieve one atom of any element from a reservoir. The difference between the two values of the number of atoms corresponding to the surface is the value of the formation energy.

**Table 1.** Parameters and associated values used for the calculations.

| Parameter | Value in Calculation |
| --- | --- |
| Exchange and correlation energy | GGA-RPBE |
| Cutoff energy | 350 eV |
| k-point mesh | $8 \times 8 \times 1$ |
| Force tolerance | 0.001 eV/Å |
| Basis set | DZP |
| DM tolerance | 0.0001 |
| SCF iterations | 1000 |
| Geometry optimization | cg |

## 3. Results and Discussion

Figure 2 shows the 3C-SiC (1 1 0) monolayers with different passivation schemes after complete geometry optimization. A comprehensive review of all bond lengths and internal angles in all structures was then performed. It is found that the structure that undergoes the most structural changes in its interior angles is the original structure, which has different angles with $\alpha = 113.74°$, $\beta = 113.7°$ and $\gamma = 89.72°$. These results are consistent with those of Bernd et al. [22] who found that the C surface atoms are displaced outward and the Si surface atoms are displaced inward. This relaxation leads to a non-negligible deformation of the first surface layer. The surface hydrogenation does not cause significant changes in the structure, the internal angles are 109.3°, while the angles formed by the hydrogen atoms bonded to the carbon and silicon atoms are 108.95° and 1 1 0.5°, similar results have been reported previously [28,29]. When the hydrogen atoms are replaced by lithium, the internal angles are 109.3° (as reported in Table 2) and the angles formed by the lithium bonded to the carbon and silicon atoms are 109.2° and 109.5°, respectively. Analysis of electron density (not shown) and Voronio charges suggests that there are no interactions other than those shown in Figure 2c, although the Li–Li, Si–Li, or C–Li interaction distances are very similar to those reported previously. In the passivation scheme CH+SiLi [Figure 2d], the internal angles show a slight decrease in their values compared to the hydrogenated case ($\alpha = 106.1°$, $\gamma = 107.4°$), while $\beta = 109.34°$ remains. In addition to the internal deformation, two different Si–Li angles are observed in this passivation (86.7° and 124.5°). These findings state that the lithium atoms lead to a deformation of the structure. Similar observations have been made experimentally in bulk Si, where lithium causes an amorphization of the structure at room temperature [30,31].

No significant changes are observed in the CLi+SiH and H+1LiSi passivation schemes, and the angles in the core remain at 109.3°, similar to the hydrogenated case. Finally, the passivations H+1LiC and H+Li$_c$ show structural changes in their internal structure, the first one has an angular value of $\alpha = 1$ 1 0.9°, $\beta = 1$ 1 0.9° and $\gamma = 104.1°$, these results indicate that lithium exerts a small compression on the structure, while in the second $\alpha = 108.1°$, $\beta = 111.9°$ and $\gamma = 108.3°$. All values of Si–C, Si–Li, Si–H, C–Li, and C–H bond lengths in each of the passivation schemes shown in Figure 2 can be found in Table 1 and Supplementary Material S1 (see Table S1 for details).

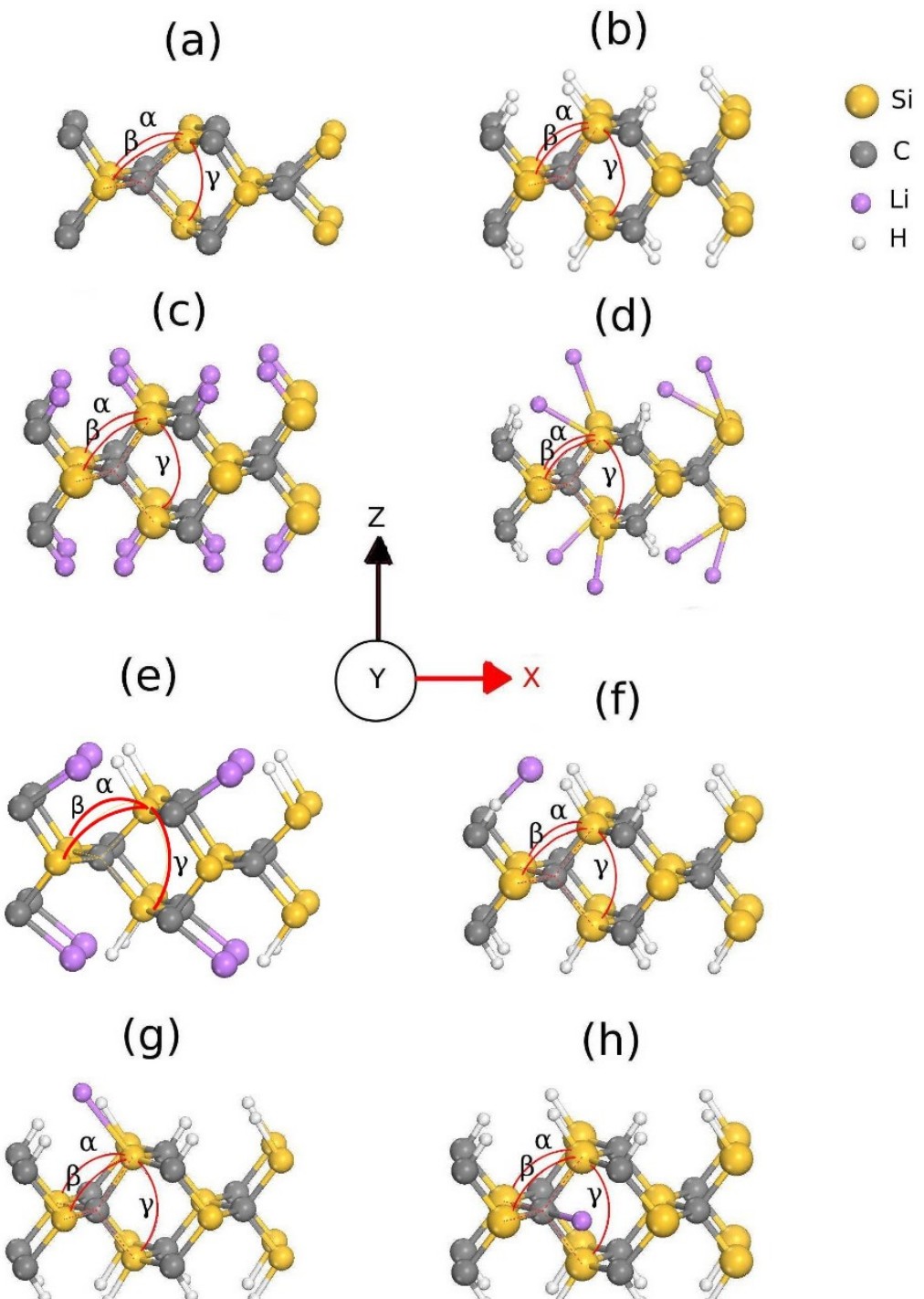

**Figure 2.** Schematic representation of 3C-SiC (1 1 0) monolayer with different passivations schemes after relaxation: (**a**) Pristine; (**b**) Full-H; (**c**) Full-Li; (**d**) CH+SiLi; (**e**) CLi+SiH; (**f**) H+1LiC; (**g**) H+1LiSi; (**h**) H+Li$_c$. The grey, yellow, white, and purple spheres denote the C, Si, H, and Li atoms, respectively.

**Table 2.** Bond lengths between atoms involved in close interaction Si–C, Si–H, Si–Li, C–H, C–Li and the experimental data.

| Passivation Scheme | Angles (°) | Bond Length (Å) | Experimental Value (Å) |
|---|---|---|---|
| Pristine | $\alpha$ = 113.74<br>$\beta$ = 113.7<br>$\gamma$ = 89.72 | Si–C = 1.91 | |
| Full-H | $\alpha$ = 109.4<br>$\beta$ = 109.4<br>$\gamma$ = 109.7 | Si–C = 1.89<br>Si–H = 1.52<br>C–H = 1.11 | |
| Full-Li | $\alpha$ = 109.6<br>$\beta$ = 109.6<br>$\gamma$ = 108.9 | Si–C = 1.89<br>Si–Li = 1.58<br>C–Li = 1.19 | |
| CH+SiLi | $\alpha$ = 106.11<br>$\beta$ = 109.34<br>$\gamma$ = 107.46 | Si–C = 1.95<br>Si–Li = 2.56<br>C–H = 1.12 | |
| CLi+SiH | $\alpha$ = 109.6<br>$\beta$ = 109.6<br>$\gamma$ = 109.1 | Si–C = 1.8, 1.9<br>Si–H = 1.65<br>C–Li = 2.07 | Si–C = 1.93 [32]<br>Si–H = 1.89 [33]<br>Si–Li = 2.57–3.09 [34,35]<br>C–H = 1.08–1.10 [32,36]<br>C–Li = 2.02 [37] |
| H+1LiC | $\alpha$ = 109.9<br>$\beta$ = 1 1 0.9<br>$\gamma$ = 104.11 | Si–C = 1.92<br>Si–H = 1.56<br>C–H = 1.11<br>C–Li = 2.1 | |
| H+1LiSi | $\alpha$ = 109.1<br>$\beta$ = 109.1<br>$\gamma$ = 108.3 | Si–C = 1.89<br>Si–H = 1.54<br>Si–Li = 2.54<br>C–H = 1.11 | |
| H+Li$_c$ | $\alpha$ = 108.1<br>$\beta$ = 111.9<br>$\gamma$ = 108.3 | Si–C = 1.96<br>Si–H = 1.51<br>C–H = 1.11<br>C–Li = 1.93 | |

Figure 3 reveals the electronic band structure of the 3C-SiC (1 1 0) surfaces with different passivation systems.

It can be observed that with Full-H passivation, the band-gap opens and becomes direct with a value of $E_{gap}$ = 2.9 eV (Figure 3b). On the other hand, Pristine, Full-Li, CH+SiLi and H+Li$_c$ (Figure 3a,c,d,h) produces a metallic behavior. The passivation scheme CH+SiLi Figure 3d causes an indirect decrease in the band-gap energy with a value of 0.87 eV. Finally, an almost flat state below the Fermi level is observed in the H+1LiC and H+1LiSi passivations, similar to the states produced by p-doping. These states have already been observed in SiC nanowires [28].

Figure 4 provides information on the partial density of states (PDOS) of the 3C-SiC (1 1 0) surfaces. It can be seen that the band-gap opens in the case of Full-H with some levels corresponding to H, C and Si atoms. It is worth noting that this behavior has been observed before and is due to charge transfer from Si to C [28].

In contrast, in the case of Full-Li and CH+SiLi, the Li interferes with the charge transfer between Si and C, allowing the free electrons to move freely across the surface, indicating metallic behavior. When the Li atoms are connected to the C atoms, the mobile electrons correspond to the Li atoms.

Figure 5 shows the formation energy versus the passivation scheme considered in this work. In the pristine monolayer, we have unpassivated atoms on the surface, which means that it is an unstable structure. Since the surface-to-volume ratio is high, it is also possible that the final energy depends on the reactivity of the atoms on the surface, which

could have some interactions due to the unpassivation, which is the reason for a positive value of this energy. We confirm that for the full-H monolayer, where we passivated all dangling bonds with H atoms, the surface energy decreases and the value of the formation energy also decreases. By this, we found that the passivation schemes Full-H is the most stable configuration followed by H+1LiC, H+1LiSi, H+Li$_c$ and CLi+SiH due to they show negative formation energy, with values of $-0.22$, $-0.2$, $-0.14$ and $-0.02$ eV, respectively. While the most unstable configurations are the pristine, Full-Li and CH-SiLi with positive formation energy with values 0.65 eV, 0.24 eV and 0.04 eV, respectively, Supplementary Material S1 (see Table S1 and Figure S1 for details). These results suggest that lithium affects the stability of the system, especially if lithium is linked to silicon atoms.

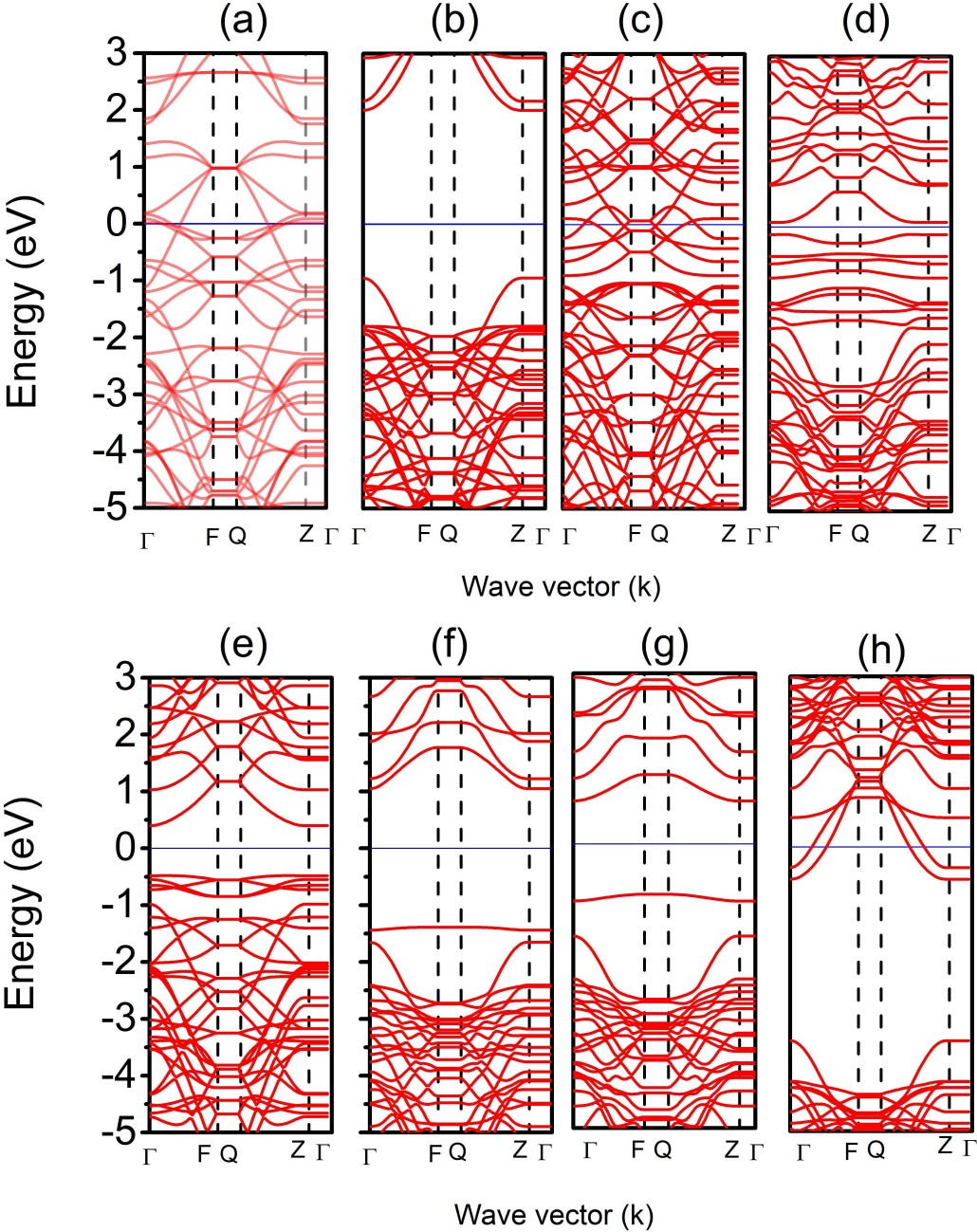

**Figure 3.** Electronic band structure of symmetric 3C-SiC (1 1 0) surfaces. (**a**) Pristine, (**b**) Full-H, (**c**) Full-Li, (**d**) CH+SiLi, (**e**) CLi+SiH, (**f**) H+1LiC, (**g**) H+1LiSi, (**h**) H+Li$_c$.

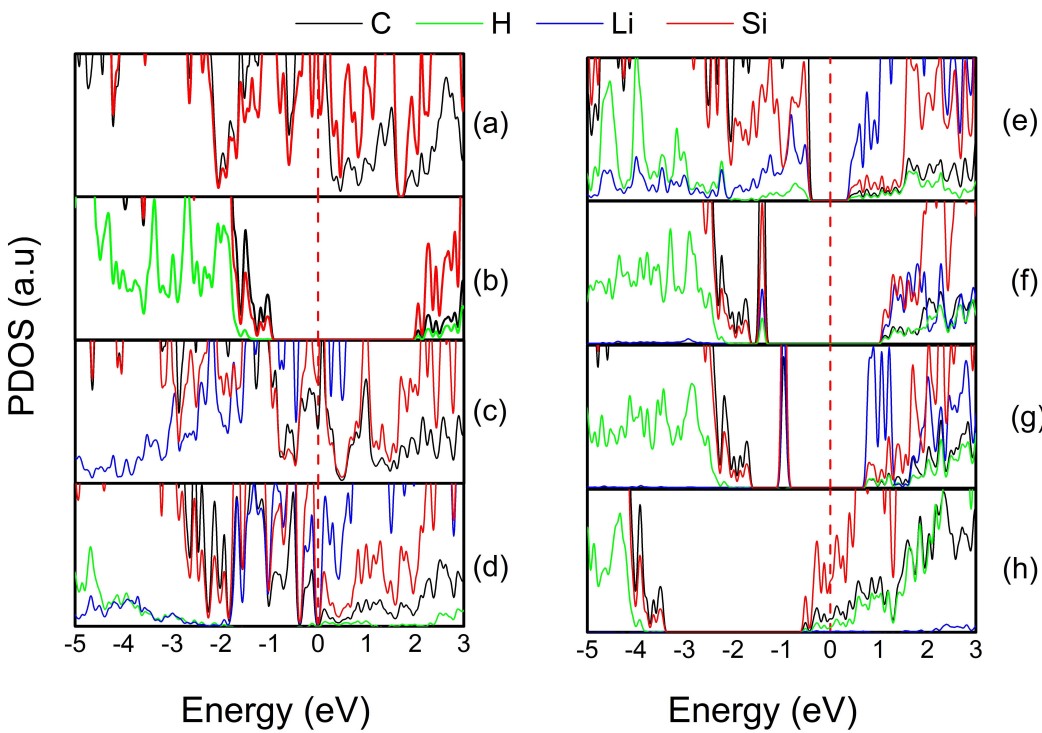

**Figure 4.** Partial Density of States of 3C-SiC (1 1 0) surfaces. (**a**) Pristine, (**b**) Full-H, (**c**) Full-Li, (**d**) CH+SiLi, (**e**) CLi+SiH, (**f**) H+1LiC, (**g**) H+1LiSi, (**h**) H+Li$_c$.

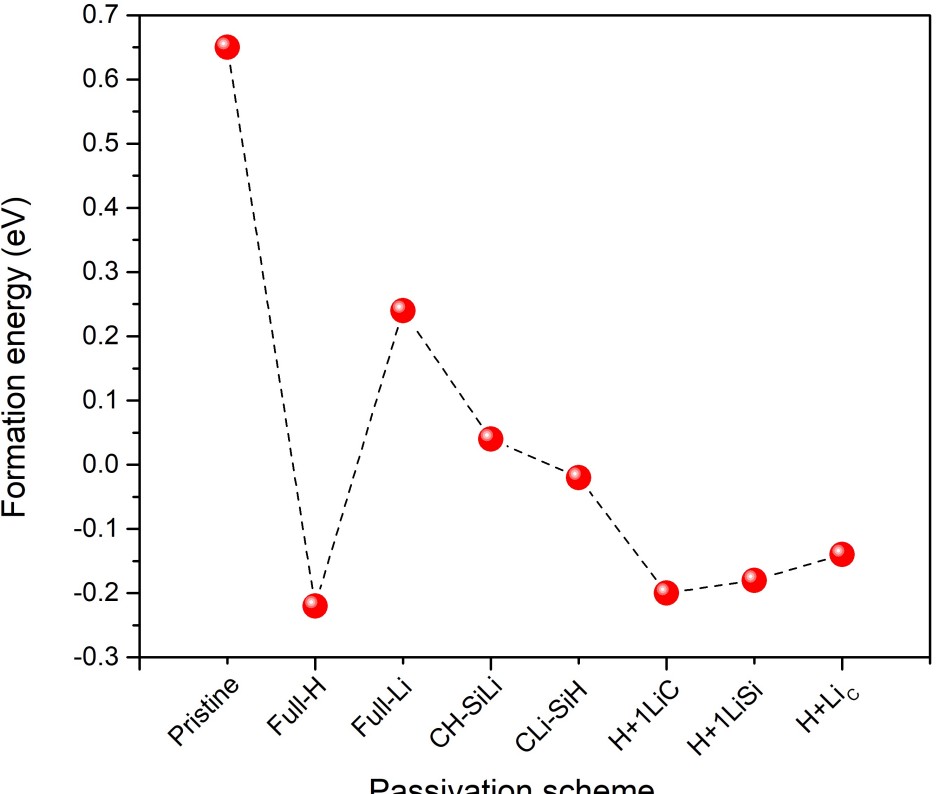

**Figure 5.** Formation energy of 3C-SiC(1 1 0) surfaces.

## 4. Conclusions

In this work, we conducted a study on the effects of lithium on the electronic properties of 3C-SiC (1 1 0) surfaces in the context of DFT-GGA. The passivation schemes Full-Li,

CH+SiLi and H+Li$_c$ have a metallic behavior, while the passivations CLi+SiH, H+1LiC and H+1LiSi decrease the band-gap and exhibit an indirect band-gap. These effects could be related to the differences in electronegativity of the individual passivating species, such that a species with higher electronegativity produces a smaller energy gap. On the other hand, full-H passivation and Full-Li passivation are the most stable and unstable systems, respectively. However, this stability can be compromised when lithium is bonded to silicon or carbon atoms. Lithium produces larger formation energies (negative) when bonded to carbon atoms, while it produces smaller formation energies (positive) when bonded to silicon atoms.

**Supplementary Materials:** The following supporting information can be downloaded at: https://www.mdpi.com/article/10.3390/batteries8110247/s1, Figure S1: Schematic representation of 3C-SiC (1 1 0) with surfaces different passivations schemes after relaxation (a) Pristine, (b) Full-H, (c) Full-Li, (d) CH+SiLi, (e) CLi+SiH, (f) H+1LiC, (g) H+1LiSi, (h) H+Li$_c$. The grey, yellow, white and purple indicate the C, Si, H and Li atoms, respectively, Table S1: Bond lengths between atoms involved in close interaction Si–C, Si–H, Si–Li, C–H, C–Li and the experimental values, Figure S2: Variation of the final energy in dependency of the cutoff energy for 3C-SiC, Figure S3: Variation of the final energy in dependency of the k-point mesh for 3C-SiC.

**Author Contributions:** Conceptualisation, S.P.T., J.L.C and M.O.M.; methodology, M.O.M. and J.L.C.; software, J.L.C., M.O.M. and S.P.T.; validation, J.L.C. and S.P.T.; formal analysis, M.O.M. and J.L.C.; investigation, S.P.T. and M.O.M.; resources, M.O.M. and S.P.T.; data curation, J.L.C., S.P.T. and M.O.M.; writing—original draft preparation, S.P.T., M.O.M. and J.L.C.; writing—review and editing, J.L.C. and S.P.T.; visualization, M.O.M. and S.P.T.; supervision, S.P.T.; project administration, S.P.T.; funding acquisition S.P.T. All authors have read and agreed to the published version of the manuscript.

**Funding:** Financial support from the seed grant "Computational modeling of biomaterials and applications to bioengineering and classical and quantum machine learning for predicting social engineering (2022–2026, code: INV-0012-042)", Universidad Indoamérica, Ecuador, awarded to S.P.T.

**Institutional Review Board Statement:** Not applicable.

**Informed Consent Statement:** Not applicable.

**Data Availability Statement:** Not applicable.

**Conflicts of Interest:** The authors declare no conflict of interest.

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
