# Peer review of "Band-Gap Engineering: Lithium Effect on the Electronic Properties of Hydrogenated 3C-SiC (1 1 0) Surfaces"

_batteries, doi:10.3390/batteries8110247_

Round 1

Reviewer 1 Report

Referee report on “Band-gap engineering: Lithium effect on the electronic properties of hydrogenated 3C-SiC (1 1 0) surface" by Jose Luis Cuevas al.

Although this topic is of some interest, this manuscript in its present form cannot be recommended for publication and requires significant revision.

1.     Line 30-32. This sentence needs supporting references. This is important to attract more reader interest and further incentive applications. For some of them, see, for example:

a) Huczko, A., Dąbrowska, A., et al . Silicon carbide nanowires: synthesis and cathodoluminescence. physica status solidi (b), 2009, 246(11‐12), 2806-2808.

b) Sameera, J.N., Islam, M.A., Islam, S., Hossain, T., Sobayel, M.K., Akhtaruzzaman, M., Amin, N. and Rashid, M.J., 2022. Cubic Silicon Carbide (3C–SiC) as a buffer layer for high efficiency and highly stable CdTe solar cell. Optical Materials123, p.111911.

c) Calabretta, C.; Scuderi, V.; Anzalone, R.; Mauceri, M.; Crippa, D.; Cannizzaro, A.; Boninelli, S.; La Via, F. Effect of Nitrogen and Aluminum Doping on 3C-SiC Heteroepitaxial Layers Grown on 4° Off-Axis Si (100). Materials 202114, 4400.

2.     The literature review in the Introduction does not mention whether there are comparative studies of the two surfaces <100> and <110>.  Why?

3.     Line 33. Sentence “In this sense, many studies have been performed on 3C-SiC(100) surfaces [9–12]…”.  It is important to note, that these references are 2003, 2005, 2007, 2002.  Does this mean that after 2007 this topic became uninteresting or the authors simply do not know new works? This is important for understanding the relevance and prospects of this work.

4.     Why is there a calculation for only one type of surface in the work?

5.     Line 77. "A comprehensive review of all bond lengths and interior

angles was then performed on all structures " Can this data be given in more detail and brought into the Supplement

 In general, the manuscript is interesting and can be recommended for publication after constructive reflection on the above comments.

Author Response

Dear Reviewer, Batteries, MDPI

Kind greetings

Thank you for considering our manuscript entitled " Band-gap engineering: Lithium effect on the electronic properties of hydrogenated 3C-SiC (1 1 0) surface."

We are very grateful to you for giving us enough time to submit the revised manuscript. We have used this time to review the quantum chemical calculations necessary to respond to some of the reviewers' comments. In addition, we have subjected the manuscript to English proofreading and corrected the grammatical and structural form of the same.

    1.  

We have changed the tittle to “Band-gap engineering: Lithium effect on the electronic properties of hydrogenated (1 1 0) surface of 3C-SiC monolayer

    1.  

We have also included supplementary material to support our findings.

All changes are colored blue in the revised manuscript. In addition, orange font is a new marker beginning with a letter in parentheses "(R. #. #)" indicates the answer to the reviewer's questions.

Reply

    1.  

Referee report on “Band-gap engineering: Lithium effect on the electronic properties of 1. hydrogenated 3C-SiC (1 1 0) surface" by Jose Luis Cuevas al.

Although this topic is of some interest, this manuscript in its present form cannot be recommended for publication and requires significant revision. 1. Line 30-32. This sentence needs supporting references. This is important to attract more reader interest and further incentive applications. For some of them, see, for example:

a) Huczko, A., Dąbrowska, A., et al . Silicon carbide nanowires: synthesis and cathodoluminescence. physica status solidi (b), 2009, 246(11‐12), 2806-2808.

b) Sameera, J.N., Islam, M.A., Islam, S., Hossain, T., Sobayel, M.K., Akhtaruzzaman, M., Amin, N. and Rashid, M.J., 2022. Cubic Silicon Carbide (3C–SiC) as a buffer layer for high efficiency and highly stable CdTe solar cell. Optical Materials123, p.111911.

c) Calabretta, C.; Scuderi, V.; Anzalone, R.; Mauceri, M.; Crippa, D.; Cannizzaro, A.; Boninelli, S.; La Via, F. Effect of Nitrogen and Aluminum Doping on 3C-SiC Heteroepitaxial Layers Grown on 4° Off-Axis Si (100). Materials 202114, 4400.

Response:

We thank you for your comments. These references were added as number 13 and 14 in the revised manuscript

2. The literature review in the Introduction does not mention whether there are comparative studies of the two surfaces <100> and <110>.  Why?

Response:

We appreciate this comment of the referee. As we found in the literature, there are many papers dealing with the differences between the (100), (111) and (110) surfaces. We have added some relevant references to improve the novelty of the work. In particular, we have added references in which the authors show that the (110) surface has the highest sensitivity to bonding with other elements such as O due to the number of silicon and carbon atoms on the (110) surface.

The following text has been added to the appropriate section of the revised manuscript,

"As we know, some researches focus on studying the different properties obtained by using (100), (111) and (110). In particular, Taehoon Park et. al [24]. show that the (110) surface is the most sensitive to oxidation compared to the others, which is due to the number of silicon and carbon atoms that are on the surface because of the surface orientation."

3. Line 33. Sentence “In this sense, many studies have been performed on 3C-SiC(100) surfaces [9–12]…”.  It is important to note, that these references are 2003, 2005, 2007, 2002.  Does this mean that after 2007 this topic became uninteresting or the authors simply do not know new works? This is important for understanding the relevance and prospects of this work.

Response:

We thank the reviewer for the comment. As we stated in the previous comment, we are adding some relevant information to improve the novelty of the work.

4. Why is there a calculation for only one type of surface in the work? The author have been working in (110) directions in SiC nanowires, however the lithium effect in the other directions will be considered in the next research.

Response:

In a previous study by the authors, (110)-SiC nanowires were investigated. For this reason, we decided to study the lithium effects on (110) monolayers, but as a next step, it could be very important to compare the lithium effects on other surfaces, but this could be done in our future study.

5. Line 77. "A comprehensive review of all bond lengths and interior angles was then performed on all structures " Can this data be given in more detail and brought into the Supplement  

Response:

Thank you for your excellent suggestion, which actually helped us to improve the details of our work, the bond length as well as the convergence tests were added as supplementary material.

We believe that the corrections have improved the quality of our work for which we are grateful.

Sincerely,

Dr. Saravana Prakash Thirumuruganandham

(https://orcid.org/0000-0003-4210-1363)

Reviewer 2 Report

The effect of lithium on the surface electronic properties of hydrogenated 3C-SiC (110) is studied in this paper, which provides useful theoretical guidance for the application of SiC in lithium-ion batteries in the future. I suggest to accept this manuscript after major revision by solving all of issues as the followings.

1. Need more concise grammar.

2. With the model size described by the author, why is 8*8*1 Kmesh adopted? Please explain the reason.

3. The setting of calculation conditions including the processing of exchange correlation energy, the selection of pseudopotential, convergence criterion and iteration step are very important. It is suggested that the authors list in the table and summarize.

4. Please briefly state the calculation method of formation energy in the calculation method section.

5. Why the formation of SiC original structure can be positive is inconsistent with common sense, please explain the reason.

6. It is suggested that the corresponding colors of elements should be marked directly in the model diagram, rather than described in the text part, so as to facilitate readers' identification.

7. The introduction of background is not enough. The author should cite more references to compare the performances. There is a recently published paper. Hopefully it can help with the improvement of the paper (ACS Nano, 2021, 15, 17232-17246; ACS Nano, 2022, 16, 9667-9678. Adv. Mater., 2022, 34, 2202877).

Author Response

Dear Reviewer, Batteries, MDPI

Kind greetings

Thank you for considering our manuscript entitled " Band-gap engineering: Lithium effect on the electronic properties of hydrogenated 3C-SiC (1 1 0) surface."

We are very grateful to you for giving us enough time to submit the revised manuscript. We have used this time to review the quantum chemical calculations necessary to respond to some of the reviewers' comments. In addition, we have subjected the manuscript to English proofreading and corrected the grammatical and structural form of the same.

    1.  

We have changed the tittle to “Band-gap engineering: Lithium effect on the electronic properties of hydrogenated (1 1 0) surface of 3C-SiC monolayer

    1.  

We have also included supplementary material to support our findings.

All changes are colored blue in the revised manuscript. In addition, orange font is a new marker beginning with a letter in parentheses "(R. #. #)" indicates the answer to the reviewer's questions.

Reply

1. The effect of lithium on the surface electronic properties of hydrogenated 3C-SiC (110) is studied in this paper, which provides useful theoretical guidance for the application of SiC in lithium-ion batteries in the future. I suggest to accept this manuscript after major revision by solving all of issues as the followings. 1. Need more concise grammar.

Response:

We have thoroughly revised the manuscript and corrected all grammatical errors

2. With the model size described by the author, why is 8*8*1 Kmesh adopted? Please explain the reason.

Response:

We do not think we clarify this point in the manuscript; we thank the reviewer for pointing it out. In the case of a periodic system, the calculations performed by DFT involve the use of a finite number of points in the first Brillouin zone, the so-called k-point mesh. The total energy depends on the choice of this k-point mesh, so it is important to use a dense value of the k-point mesh to ensure convergence of the results. In our case, we have modeled some monolayers that are small but symmetric only in x and y. This is the case for the k-point mesh. This means that a k-point mesh is needed only for this crystallographic direction, even if there is no symmetry in z. The value of 8 for the k-point mesh was obtained after some convergence tests for the crystal SiC. The convergence test is included in the supplementary material.

We included the text as an appropriate inclusion in the revised manuscript,

For the charge density integration we have used a 8 x 8 x 1 k-point Monkhorst Pack mesh. All atoms were allowed to relax until the forces were smaller than 0.001 eV/ ̊A. The main parameters employed during calculations are summarized in Table 1. To ensure the convergence of the results, we performed some convergence tests for the k-point mesh and the energy cutoff both of them are presented in the supplentary material SX.”

3. The setting of calculation conditions including the processing of exchange correlation energy, the selection of pseudopotential, convergence criterion and iteration step are very important. It is suggested that the authors list in the table and summarize.

Response:

We agree with the reviewer's suggestion and add Table 1, which summarizes the parameters used in the calculations,

4. Please briefly state the calculation method of formation energy in the calculation method section.

Response:

We have added the following text at the appropriate place in the revised manuscript, "The energy formation was calculated using equation 1. First, we relaxed all SiC monolayers, considering in each case that the resulting value of the final energy corresponds to E0. We also added the values of the binding energy per atom corresponding to the whole structure. This value represents the energy required to retrieve one atom of any element from a reservoir. The difference of the two values between the number of atoms corresponding to the surface is the value of the formation energy”

5. Why the formation of SiC original structure can be positive is inconsistent with common sense, please explain the reason.

Response:

We are very pleased with the referee's comment; we need to clarify this point. To answer this question, we must consider the SiC in the bulk, which is the origin of all the structures reported in this paper. In this sense, it could have an energy formation close to 0, since it is the reference for the binding energy of Si and C. However, in the bare monolayer we have unpassivated atoms on the surface, which means that it is an unstable structure. Moreover, due to the high surface to volume ratio, it is possible that the final energy depends on the reactivity of the atoms on the surface, which could have some interactions due to unpassivation, which is the reason for a positive value of this energy. We confirm that for the full-H monolayer, where we passivated all dangling bonds with H atoms, the surface energy decreases and the value of the formation energy also decreases.

6. It is suggested that the corresponding colors of elements should be marked directly in the model diagram, rather than described in the text part, so as to facilitate readers' identification.

Response:

Thank you for the suggestions. The changes were made in figures 1 and 2

7. The introduction of background is not enough. The author should cite more references to compare the performances. There is a recently published paper. Hopefully it can help with the improvement of the paper (ACS Nano, 2021, 15, 17232-17246; ACS Nano, 2022, 16, 9667-9678. Adv. Mater., 2022, 34, 2202877).

Response:

Thank you for your suggestions. These changes have been incorporated into the revised version of the manuscript.

We believe that the corrections have improved the quality of our work for which we are grateful.

Sincerely,

Dr. Saravana Prakash Thirumuruganandham

(https://orcid.org/0000-0003-4210-1363)

Reviewer 3 Report

The paper reports a computational investigation of 3C-SiC (110) surfaces terminated by different combinations of H and Li atoms. The paper is a little bit simplistic, in my opinion.

The main weakness I see is the choice of a very short length of the lattice parameters (Lines 47-49), as 3C-SiC has a cubic structure with lattice parameter of a=4.3596 A. Moreover a single layer of SiC seems not a proper choice to investigate the electronic properties. Can the authors justify this choice?

The authors refer to their study as an investigation of the SiC surface, but actually they are studying a 2D material or a  slab, because there is no bulk and a surface, but a single layer of SiC. Also the passivation with different atoms (H or Li) is performed on these layers (on both sides).

Figure 5: in many cases the formation energy of different surfaces is positive, which means that the surfaces are not stable. Can the authors comment on that?

Bibliography is not cited in order. Moreover, in the list of reference many lines start with a comma. Please, revise.

I guess that the paper will benefit by a more detailed and accurate choice and justification of the proposed models.

Author Response

Dear Reviewer, Batteries, MDPI

Kind greetings

Thank you for considering our manuscript entitled " Band-gap engineering: Lithium effect on the electronic properties of hydrogenated 3C-SiC (1 1 0) surface."

We are very grateful to you for giving us enough time to submit the revised manuscript. We have used this time to review the quantum chemical calculations necessary to respond to some of the reviewers' comments. In addition, we have subjected the manuscript to English proofreading and corrected the grammatical and structural form of the same.

    1.  

We have changed the tittle to “Band-gap engineering: Lithium effect on the electronic properties of hydrogenated (1 1 0) surface of 3C-SiC monolayer

    1.  

We have also included supplementary material to support our findings.

All changes are colored blue in the revised manuscript. In addition, orange font is a new marker beginning with a letter in parentheses "(R. #. #)" indicates the answer to the reviewer's questions.

Reply

The paper reports a computational investigation of 3C-SiC (110) surfaces terminated by different combinations of H and Li atoms. The paper is a little bit simplistic, in my opinion.

Response:

We thank you for your comments and have made appropriate changes in the revised version by adding an additional table and including supplementary material to support our findings to improve the manuscript.

1. The main weakness I see is the choice of a very short length of the lattice parameters (Lines 47-49), as 3C-SiC has a cubic structure with lattice parameter of a=4.3596 A. Moreover, a single layer of SiC seems not a proper choice to investigate the electronic properties. Can the authors justify this choice?

Response:

We appreciate the reviewer's comment. To clarify this point, we would like to mention that the lattice parameter of a=4.3596 corresponds to the primal cell of a monolayer. This monolayer is modeled from a perfect SiC crystal in the (110) direction. Moreover, this monolayer is subject to symmetry constraints in the x and y axes. Thus, if we reproduce the primal cell, we obtain a large surface area in 2 spatial directions. Moreover, the electronic properties in monolayers are interesting due to the propagation of electrons around the surface corresponding to the monolayer. Here, there could be some energy dispersion or different electronic mobility as it is the case in other 2D structures like graphene.

2. The authors refer to their study as an investigation of the SiC surface, but actually they are studying a 2D material or a  slab, because there is no bulk and a surface, but a single layer of SiC. Also the passivation with different atoms (H or Li) is performed on these layers (on both sides).

Response:

We agree with the reviewer and have indicated that we are studying a monolayer where we passivate the surfaces (top and bottom) that form the monolayer.

3. Figure 5: in many cases the formation energy of different surfaces is positive, which means that the surfaces are not stable. Can the authors comment on that?

Response:

We are very pleased with the referee's comment; we need to clarify this point. To answer this question, we must consider the SiC in the bulk, which is the origin of all the structures reported in this work. In this sense, the energy formation could be close to 0, since this is the reference for the binding energy of Si and C. However, in the pristine monolayer, we have unpassivated atoms at the surface, implying that it is an unstable structure. Due to the high surface to volume ratio, it is also possible that the final energy depends on the reactivity of the atoms at the surface, which could have some interactions due to unpassivation, which is the reason for a positive value of this energy. We confirm that for the full-H monolayer, where we passivated all dangling bonds with H atoms, the surface energy decreases and the value of the formation energy also decreases.

4. Bibliography is not cited in order. Moreover, in the list of reference many lines start with a comma. Please, revise.

Response:

Thank you for your comments. The order of the bibliographical references and all other corrections have been made in the revised manuscript.

5. I guess that the paper will benefit by a more detailed and accurate choice and justification of the proposed models.

Response:

Thank you very much for your comments. We have carefully reviewed the entire manuscript, made changes in the abstract, and included additional rationale in the revised manuscript.

We believe that the corrections have improved the quality of our work for which we are grateful.

Sincerely,

Dr. Saravana Prakash Thirumuruganandham

(https://orcid.org/0000-0003-4210-1363)

Round 2

Reviewer 1 Report

The authors have significantly improved their original manuscript, so now it can be recommended for publication.

Reviewer 2 Report

The authors have properly addressed all my comments. I believe the manuscript is ready for publication.

Reviewer 3 Report

The paper can be published as is.